# Precise Analysis of Nanoparticle Size Distribution in TEM Image

**DOI:** 10.3390/mps6040063

**Published:** 2023-07-03

**Authors:** Shan Zhang, Chao Wang

**Affiliations:** 1School of Materials Science and Engineering, Institute of Materials Science and Devices, Suzhou University of Science and Technology, 99 Xuefu Road, Suzhou 215011, China; 2The Russell H. Morgan Department of Radiology and Radiological Science, The Johns Hopkins University School of Medicine, 733 N. Broadway, Baltimore, MD 21218, USA

**Keywords:** nanoparticle, size distribution, transmission electron microscope, *ImageJ*

## Abstract

As an essential characterization, size distribution is an important indicator for the synthesis, optimization, and application of nanoparticles. Electron microscopes such as transmission electron microscopes (TEMs) are commonly utilized to collect size information on nanoparticles. However, the current popular statistical method of manually measuring large particles one by one, using a ruler tool in the corresponding image analysis software is time-consuming and can introduce manual errors. Moreover, it is difficult to determine the measurement interval for irregularly shaped nanoparticles. Therefore, it is necessary to use an efficient and standard method to perform size distribution analysis of nanoparticles. In this work, we use basic *ImageJ* software (1.53 t) to analyze the size of typical silica nanoparticles in a TEM image and use *Origin* software to process the data, to obtain its accurate distribution quickly. Using it as a template, we believe that this work can provide a paradigm for the standardized analysis of nanoparticle size.

## 1. Introduction

With the development of nanotechnology, nanomaterials have been extensively studied in many fields [1], with some commercial applications already underway [2]. As one of the important characteristics of nanoparticles, particle size can greatly impact their properties and applicability [3,4]. Therefore, characterization and analysis of nanoparticle size are of fundamental importance.

Currently, there are two main methods for analyzing nanoparticle size. The first is with a dynamic light scattering (DLS) analyzer, which is used to measure the hydrodynamic size of nanoparticles [5]. However, since it does not measure true particle size, it can only be used as a reference or for pre-characterization before electron microscopy measurements are taken. The second method is microscopy analysis such as transmission electron microscopy (TEM), scanning electron microscopy (SEM), and atomic force microscopy (AFM) analysis [6]. These methods provide visual images that allow for the exact measurement of nanoparticles’ size. However, compared to DLS analysis, microscopy analysis is time-consuming and requires manual measurement of at least 50 particles for statistical analysis [7]. In addition, manual measurement can introduce deviations and different researchers may select different particle diameters during the measurement process.

To address these issues, we propose an analysis method that combines the use of the commonly used software, *ImageJ*, with *Origin* to analyze TEM images. Using SiO_2_ nanoparticles as a template, this method can provide precise particle size distribution with just a few clicks. This semi-automated approach can save time and reduce the potential for human error in manual measurement. This method has the potential to revolutionize the way nanoparticle size is analyzed and could lead to more accurate and efficient results. By combining the power of commonly used software with advanced techniques, we can achieve a level of precision that was previously unattainable.

## 2. Experimental Design

### 2.1. Materials, Devices, and Software

Tetraethyl orthosilicate (TEOS), NH_3_·H_2_O (30%), and ethanol were all purchased from Sigma Aldrich (St. Louis, MO, USA); water was obtained from a deionized water system. A magnetic stirring bar and a 250 mL beaker with a stirring station (Isotemp™ Hot Plate Stirrer, ambient to 400 °C, ceramic) were purchased from Fisher Scientific (Waltham, MA, USA). Transmission electronic microscope (TEM) images were obtained from a TEM system (Hitachi 7600 TEM, Hitachi, Tokyo, Japan) with an accelerating voltage of 75 keV. For *ImageJ* and Origin, each historic version was available; here *ImageJ* 1.53 t bundled with 64-bit and Origin 9.1 32-bit were used to analyze the size distributions from TEM images.

### 2.2. Synthesis of Silica Nanoparticles

The synthesis of SiO_2_ nanoparticles relies on the classic Stöber process [8] with some improvements: firstly, 1 mL TEOS is mixed with 9 mL ethanol to form Solution A, and 9 mL NH_3_·H_2_O is added into 90 mL ethanol to prepare Solution B. Thereafter, add 9 mL Solution A into Solution B drop by drop, keep stirring (stirring speed is 400 rpm) in room temperature for 20 min, then the mixture solution is collected and centrifuged three times with water (at 10,000 rpm, 10 min).

### 2.3. Characterization of Silica Nanoparticles

After the synthesis, the SiO_2_ nanoparticles are re-dispersed into ethanol to increase their dispersion and dropped onto the copper mesh for TEM characterization. The image with excellent dispersion is selected as the purpose sample for further analysis.

## 3. Procedure

The TEM image of SiO_2_ nanoparticles is shown in Figure 1, which shows the uniform size of SiO_2_ nanoparticles and excellent dispersion. The size of SiO_2_ nanoparticles is roughly estimated to be below 100 nm, but a more accurate value cannot be obtained through visual judgment alone.

To analyze the image, *ImageJ* software is used. Firstly, open *ImageJ* software and import the image (Figure 2A), then click on the Analyze and Set Measurements buttons in turn, to set corresponding parameters (Figure 2B). The Area option is selected as the analysis object to measure the square of particles (Figure 2C). It is worth mentioning that if your particles are irregular, you can also click the Feret’s diameter button as a parameter to obtain a solid value for particle diameter. After that, the size of the nanoparticle should be synchronously mapped into the parameter of *ImageJ*. Firstly, the photo is magnified in the scale bar section as much as possible, and a straight line is drawn with the same length as the scale bar (Figure 2D). Then, click the Analyze and Set Scale buttons in turn to set it, change the Known Distance value to the scale bar length value (it is 500 nm in this image, so we input 500) and set Unit of Length as “nm” (Figure 2E). 

This process allows for a more precise analysis of the nanoparticle size by using advanced software tools. By mapping the size of the nanoparticle into the parameter of *ImageJ*, we can achieve a level of accuracy that was previously unattainable through visual judgment alone.

After importing the image into *ImageJ* and setting the parameters, the next step is to adjust the image format and threshold value. Click the Image and Type in turn, select 8-bit as image format (Figure 3A), then click Image, Adjust, and Threshold in turn to set the threshold value (Figure 3B); usually, we use the system default value, so just click Set and OK in turn (Figure 3C). By this point, the nanoparticles are separated from the background due to their different brightness and contrast. Then, we can analyze the size distribution of silica nanoparticles by clicking on Analyze and Analyze particles in turn (Figure 3D), setting Size from 1000 (if you set zero as the value, other impurities may also be calculated and influence the results, so some the minimum value should be set to avoid the situation, depending on your trial) to infinity (Figure 3E), then click OK, obtain the results of individual area (square) and the number of the corresponding nanoparticles (Figure 3F). Here, we obtained 276 nanoparticles and their individual squares. This process allows for a more precise analysis of the size distribution of silica nanoparticles by using advanced software tools. By adjusting the image format and threshold value, we can separate the nanoparticles from the background and accurately analyze their size distribution.

After analyzing the square of silica nanoparticles using *ImageJ*, the diameter size of the nanoparticles can be further analyzed using *Origin* software. Firstly, copy individual area values to the *Origin* workbook, the numbers are set as x value, the squares are set as y value (Figure 4A), and abnormal values should be deleted (Figure 4B and its inner Figure, in the TEM image, few particles linked together and are counted as one, so this kind of particles should be removed to increase the accuracy of calculation), then, we need to obtain the diameter values from area (square) values, here we click another column in the workbook (that is Column C)and Set Column Values in turn, according to the simple square-diameter formula of round:(1)$$S=\pi r^{2}=\pi {\left(\frac{d}{2}\right)}^{2}\;d=2\times \sqrt{S/\pi }$$
set Column C as the diameter value (Figure 4C). Then, click the Column C, Plot, Statistics, and Histogram in turn (Figure 4D) to obtain a histogram of the values (Figure 4E), optimize the histogram, and obtain the results. The average diameter of particles is calculated by dividing the sum of all available particle sizes by the total number of available particles. This information is then used to generate the final diagram, as shown in Figure 4F. From the results, we can deduce that the average size of the silica nanoparticles is 70.6 nm, and it has a narrow size distribution that fits the Gauss Curve, showing its excellent distribution property. 

This process allows for a more precise analysis of the size of silica nanoparticles by using advanced software tools. By combining the power of *ImageJ* and *Origin*, we can achieve a level of accuracy that was previously unattainable through manual measurement alone.

## 4. Protocol Limitations and Further Applications

This method is most effective when used on regular and un-agglomerated particles, such as the round and dispersed silica nanoparticles in this work. By analyzing these kinds of nanoparticles, we can obtain a uniform diameter for each particle, rather than one of the specific diameters (maybe the maximum or minimum diameter) obtained manually, which may select the diameter deliberately or induce a significant error unintentionally.

For irregular particles, the Feret’s diameter setting in ImageJ (this parameter can be set in Figure 2C, below the Set Measurements button) can be used to address this issue, though it cannot provide a uniform diameter, but only a specific diameter for each particle. However, this method is not suitable for analyzing aggregated nanoparticles due to the lack of an aggregated particle discrimination function in ImageJ [9]. In the future, artificial intelligence used in image processing software may be able to solve this problem to improve the utility of this method [10].

Besides shape and distribution, contrast is another important indicator, as it decides whether the particles can be separated effectively from the background. In this case, some low-contrast particles may not be suitable, such as lipid particles or vesicles.

In addition to nanoparticle size analysis, this method can also be applied to the size analysis of other small particles or particle analogs, such as microparticles and even bacteria and cells obtained from optical microscope images; for bacteria and cell analysis, we believe that staining them is the most suitable means to distinguish them from the background in processing. Except for those used for TEM images, the images obtained from other electron microscopes such as SEM or AFM can also be analyzed by using the same method.

## 5. Conclusions

By using ImageJ for image analysis and Origin for data processing, this semi-automated method can accurately determine the average particle size and size distribution of SiO_2_ nanoparticles in just a few minutes. Compared with manual analysis by measuring the size of particles one by one, it significantly reduces the measurement time and significantly increases the accuracy of measurement. The average diameter of the silica nanoparticles was found to be 70.6 nm. The size distribution of the nanoparticles was narrow and showed excellent uniformity. This analysis method can also be used for any particle size analysis of SEM and AFM images or other bacteria and cells from an optical microscope.

This method provides a fast and accurate way to analyze the size distribution of nanoparticles and other small particles in any optical microscope and electron microscope images. By combining advanced software tools with a well-designed experimental protocol, we can achieve precise and reliable results.

## Figures and Tables

**Figure 1 mps-06-00063-f001:**
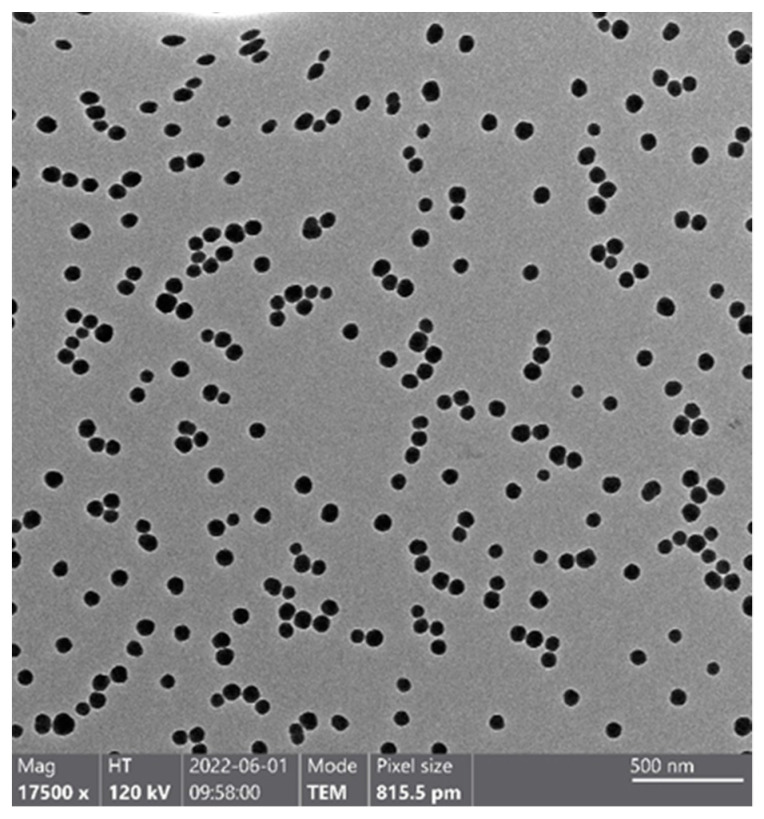
TEM image of SiO_2_ nanoparticles. It shows nanoscale, uniform (non-aggregation), and high-contrast (from background) silica nanoparticles, which is suitable for the following analysis.

**Figure 2 mps-06-00063-f002:**
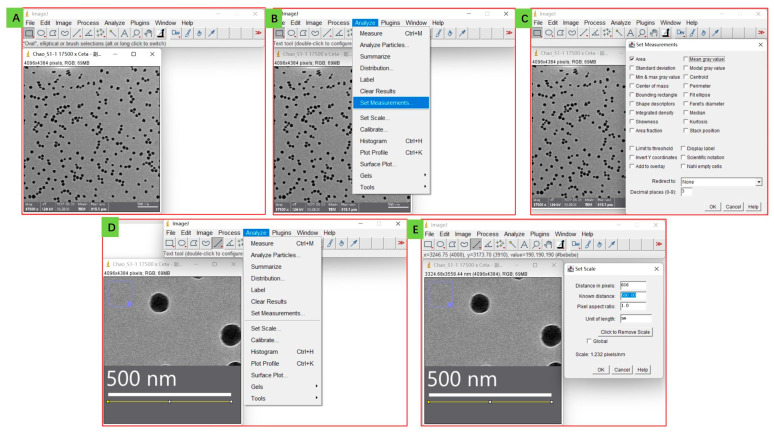
Analysis by *ImageJ* software: from importing the image (**A**) to setting measurement parameters (**B**,**C**) to calibrating the scale bar (**D**,**E**).

**Figure 3 mps-06-00063-f003:**
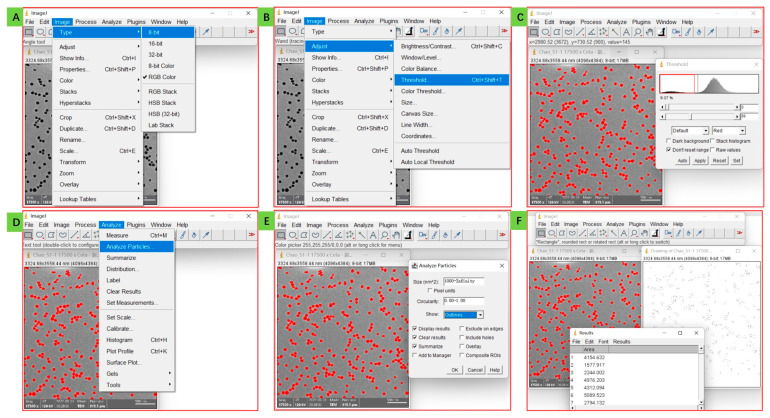
Analysis by *ImageJ* software: adjust the image format (**A**), separate nanoparticles from the background (**B**,**C**), and analyze and obtain the square values of each nanoparticle (**D**–**F**).

**Figure 4 mps-06-00063-f004:**
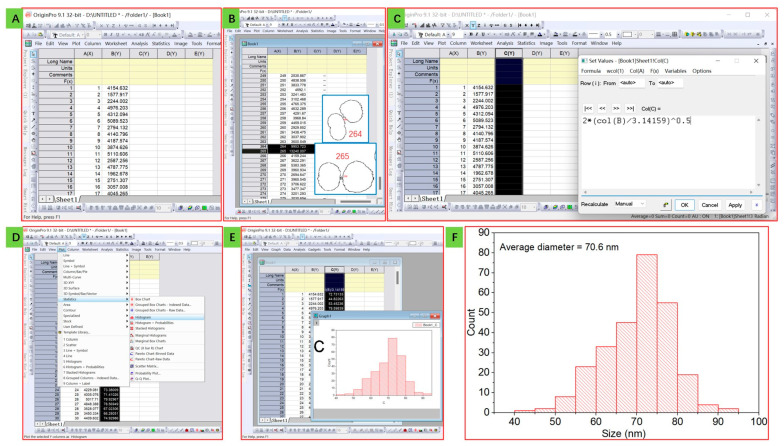
Analysis using Origin software: import the data from results of *ImageJ* (**A**), delete wrong values of linked particles (**B**), calculate the diameter using the diameter-square equation (**C**), and obtain the histogram of the precise size distribution of silica nanoparticles (**D**–**F**).

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
