# Peer review of "Precise Analysis of Nanoparticle Size Distribution in TEM Image"

_mps, 2023, doi:10.3390/mps6040063_

Round 1

Reviewer 1 Report

1.      The introductory part is way too carefree and way too short. It would be helpful if the authors discussed this in light of recent references on the subject. For more up-to-date references compared with another nanoparticle, authors should familiarize themselves with the following article:  https://doi.org/10.1016/j.apsusc.2020.145930.

2.      The work needs to be modified to include a schematic representation of the experimental method to be used to create SiO2 nanostructures. The schematic diagram of the prepared samples should be included.

3.      The authors need to check the phases of the SiO2 nanoparticles by performing an XRD pattern on the SiO2 NPs.

4.      To determine particle size, authors used TEM images with Image J software. Why didn't the authors run HRTEM and SAED patterns on the SiO2 to confirm the phases of SiO2, which should match well with the XRD results?
5. The authors should improve the resolution of all figures included in the revised manuscript.

Comments:

1.      The introductory part is way too carefree and way too short. It would be helpful if the authors discussed this in light of recent references on the subject. For more up-to-date references compared with another nanoparticle, authors should familiarize themselves with the following articles: DOI: 10.1039/C4CP05679E; https://doi.org/10.1016/j.apmt.2019.02.016;   https://doi.org/10.1002/pssa.201800768; https://doi.org/10.1016/j.apsusc.2020.145930.

2.      The work needs to be modified to include a schematic representation of the experimental method to be used to create SiO2 nanostructures. The schematic diagram of the prepared samples should be included.

3.      The authors need to check the phases of the SiO2 nanoparticles by performing an XRD pattern on the SiO2 NPs.

4.      To determine particle size, authors used TEM images with Image J software. Why didn't the authors run HRTEM and SAED patterns on the SiO2 to confirm the phases of SiO2, which should match well with the XRD results?
5. The authors should improve the resolution of all figures included in the revised manuscript

If the authors wish to address these analyses, I would suggest that the manuscript be submitted for publication to the Journal of Methods and Protocols (MDPI).

Reviewer 2 Report

The manuscript describes a precise analysis of nanoparticle size distribution in TEM images. This method is not new. There is no significant finding in this manuscript.  We can find on youtube many tutorials for size calculation by using ImageJ combine with Origin. 

In addition, the nanoparticle of SiO2 is quite easy to be determined their size through this method. This method is useless to be applied for size calculation on irregular particles, especially agglomerated particles. 

Minor editing of English language required

Reviewer 3 Report

The article entitled “Precise analysis of nanoparticle size distribution in TEM image” is clearly presented. The people who are all working on the materials science would know these procedures. More details available in the documentation of ImageJ software. However, these clear instructions through step by step images are good for easy understanding by the new or upcoming students and researchers. Hence I recommend this may be considered for publication.

 The average diameter measurement (Figure 4F) is not discussed.

The section 5 is there without any content. The authors may remove that or should add the patent details.

Reviewer 4 Report

The protocol described is very useful for the specialists on the field. The manuscript is well organised and presented. The conclusions are sustained by the obtained results. It is a great protocol for TEM images. Several aspects need attention according to the comment below:

Comment 1) Line 87: I recommend to remove the expression ,,size analysis of SEM and AFM images” from conclusions. It should be placed and discussed in a new paragraph entitled ,,Protocol limitations and further applications”

Comment 2) I use Image J coupled with Origin for the analysis of SEM and AFM images using a semi-automated protocol regarding particles diameter measuring. There are a lot of agglomerated particles in SEM and AFM images that significantly limits the successfully application of the protocol as you described and it might affect the resulted value. Therefore, it further requires a lot of computational programming behind the threshold setting in Image J to have a real success for SEM and AFM images on the full automated mode. I wish you a lot of success on the future improvements of the protocol.

Comment 3) Some newest references (2020 – 2023) should be added to the discussion.

Minor editing of English language required.

Round 2

Reviewer 2 Report

The manuscript has been revised accordingly. 

The manuscript has been revised accordingly. 

Author Response

Thank you for your agreement.